# Molecular Simulation on Competitive Adsorption Differences of Gas with Different Pore Sizes in Coal

**DOI:** 10.3390/molecules27051594

**Published:** 2022-02-28

**Authors:** Qing Han, Cunbao Deng, Tao Gao, Zhixin Jin

**Affiliations:** College of Safety and Emergency Management and Engineering, Taiyuan University of Technology, Jinzhong 030600, China; dengcunbao@tyut.edu.cn (C.D.); gaotao0174@link.tyut.edu.cn (T.G.); jinzhixin@tyut.edu.cn (Z.J.)

**Keywords:** coal, competitive adsorption, GCMC, pore size, enhanced coalbed methane recovery

## Abstract

Micropores are the primary sites for methane occurrence in coal. Studying the regularity of methane occurrence in micropores is significant for targeted displacement and other yield-increasing measures in the future. This study used simplified graphene sheets as pore walls to construct coal-structural models with pore sizes of 1 nm, 2 nm, and 4 nm. Based on the Grand Canonical Monte Carlo (GCMC) and molecular dynamics theory, we simulated the adsorption characteristics of methane in pores of different sizes. The results showed that the adsorption capacity was positively correlated with the pore size for pure gas adsorption. The adsorption capacity increased with pressure and pore size for competitive adsorption of binary mixtures in pores. As the average isosteric heat decreased, the interaction between the gas and the pore wall weakened, and the desorption amount of CH_4_ decreased. In ultramicropores, the high concentration of CO_2_ (50–70%) is more conducive to CH_4_ desorption; however, when the CO_2_ concentration is greater than 70%, the corresponding CH_4_ adsorption amount is meager, and the selected adsorption coefficient S_CO__2/CH__4_ is small. Therefore, to achieve effective desorption of methane in coal micropores, relatively low pressure (4–6 MPa) and a relatively low CO_2_ concentration (50–70%) should be selected in the process of increasing methane production by CO_2_ injection in later stages. These research results provide theoretical support for gas injection to promote CH_4_ desorption in coal pores and to increase yield.

## 1. Introduction

With the increase in coal mining depth, the corresponding incidence of mine gas disasters shows an increasing trend. Therefore, mine gas prevention has become a critical factor for achieving safe and efficient production in mines [1,2,3]. According to BP 2021, in 2020, coal accounted for 27.2% of the world’s primary energy consumption and was the world’s second-leading energy source [4]. At present, most production processes have tended in the direction of developing clean energy. The existing data show that China’s total natural gas production in 2020 was about 1876.44 billion m^3^. The total production of coalbed methane was about 20.4 billion m^3^ and the production of tight gas was about 4.4 billion m^3^. The total production of coal-measure gas was about 24.8 billion m^3^ [5]. Compared with developed countries such as the United States, China’s CBM production remains low, and its external dependence remains high. However, China is rich in coalbed methane reserves. According to the preliminary prediction, China’s coalbed methane reserves of less than 2000 m could reach 36.18 trillion m^3^ [5].

Coal is heterogeneous and has a large number of pore structures. Due to the uneven distribution of pores in coal with different metamorphic degrees, different pore structures lead to differences in the adsorption, desorption, diffusion, and migration of coalbed methane. The occurrence states of gas in coal are mainly the adsorption state and the free state, and the principal occurrence area is the coal pore structure. The existing research shows that the adsorbed gas in coal pores can reach 80% and above [6,7,8]. Many researchers have studied the pore size distribution of coal based on experimental measurements and other methods. Coal has been comprehensively studied as a porous medium using constantly updated methods in recent years, including SEM, TEM, MIP, NMR, CT, FIB-SEM, AFM, and low-temperature nitrogen and carbon dioxide adsorption/desorption [9,10,11]. Different methods have generally different detectable scales, and a combination of different experimental technologies is necessary for a systematic illustration of the different pore distributions and pore structure variations. IUPAC divides coal pore size into micropores, mesopores, and macropores [12]. The leading gas adsorption sites are micropores [13,14], and seepage takes place mainly through macropores. At the same time, due to the close distance between the pore walls in the micropores, there is a strong intermolecular interaction between the methane adsorbed in the pores and the pore walls. As a result, the methane molecules in the adsorbed state must change from the adsorbed state to the free state under certain external conditions. Therefore, the methane present in micropores is difficult to extract [15]. In addition, competitive adsorption is often carried out by injecting high-pressure CO_2_ or N_2_, or by using other methods to realize the displacement of CH_4_ and, thus, effectively improve the recovery rate of CBM. Song et al., taking low-rank coal as the research object, studied the competitive adsorption characteristics of CH_4_ and CO_2_ through experiments and simulation methods, and they found that vitrinite in coal had a strong adsorption capacity for CO_2_ [16]. Shan et al. [17], Wu et al. [18], and Zhou et al. [19] studied the difference in adsorption characteristics of power plant flue gas in a coal matrix through experiments and molecular simulations.

Researchers have carried out relevant studies on methane’s occurrence and flow state in pores by molecular simulation. Wang et al. reviewed the recent molecular simulation research on gas adsorption/desorption and diffusion in the shale matrix, which was based on DFT, GCMC, and MD methods [20]. Due to the complexity of coal’s structure, researchers replaced the coal pore structure with graphene or carbon nanotubes to further explore the influence of the external environment on the gas adsorption state in pores. Wang et al. summarized the shale molecular models and analyzed the microscopic mechanism of shale gas adsorption in detail [21]. Lin et al. simplified shale pores as graphene sheets and conducted a CH_4_/CO_2_ adsorption simulation, and their simulation results were close to the experimental results, which proved the correctness of the model [22]. Based on the GCMC method, Song et al. analyzed the methane adsorption amount in graphite micropores under different temperatures and pressures. The results showed that the excess adsorption isotherms and the maximum excess adsorption densities of different pore structures varied. The adsorption density corresponding to triangular pores was the most significant [23]. George et al. conducted a GCMC simulation of CH_4_, N_2_, CO, and CO_2_ adsorption capacity in single-wall carbon nanotubes (SWCNTs) at 298 K and 0.01–2.0 MPa, and found that the adsorption uptake of the examined SWCNTs was H_2_ << N_2_ ≈ CH_4_ < CO << CO_2_, which corresponded to the experimental results [24]. Mosher et al. simulated the competitive adsorption of CH_4_ and CO_2_ in the microporous and mesoporous structures of coal. The results showed that pressure had an important influence on the adsorption of pore gas [25]. Wu et al. explored the adsorption and replacement mechanisms of methane in carbon nanochannels using the molecular dynamics method, finding that CO_2_ could directly replace the adsorbed CH_4_, while N_2_ reduced the partial pressure of CH_4_ [15]. Bai et al. constructed a coal pore-structure model based on the coal molecular model, and investigated the desorption characteristics of CH_4_ after CO_2_ and N_2_ injection at different temperatures. The results showed that the effect of CO_2_ injection on promoting methane desorption was better than that of N_2_ [26]. Dong et al. took coal samples with different metamorphic degrees as their research objects, and compared and analyzed the adsorption differences of CH_4_ and CO_2_ with corresponding different functional groups under different coal pore parameters, revealing that the contributions of the pore structure and functional groups to CO_2_/CH_4_ adsorption were 83% and 17%, respectively [27]. Thus, the pore structure is the main gas occurrence area, and the distribution in the coal matrix has an important influence on gas adsorption.

Based on the comprehensive analysis of the existing pore structure studies, most studies, such as Lin et al. [22], Liu et al. [28], Pongtorn et al. [29], Xiong et al. [30], and so on, mainly focused on the thermodynamic changes of CH_4_ under single-layer adsorption conditions. The ranges of temperature and pressure were relatively wide. At the same time, the main problem in the studies was the adsorption of single-component gas. However, in the actual production process in coal mines, the gas adsorbed in the pores of the coal body must rely on external means to achieve its desorption. The analysis of the competitive adsorption mechanism based on coal pore structure is of great significance for the realization of efficient methane extraction in later stages. Although Song et al. explored the adsorption of porous methane, they mainly studied the differences in adsorption characteristics of different shapes of pores [23]. At the same time, the environmental conditions for the study were high-temperature and high-pressure, which were quite different from the main purpose of this study. Based on the complexity of the structure of coal, this study selected different pore sizes as the research object based on the main site micropores of methane adsorption. At the same time, the original single-layer graphene pore-wall structure was optimized by referring to the existing research results, and the three-layer, more stable graphene structure was changed to study the difference in competitive gas adsorption corresponding to different pore sizes. This research content has not appeared previously in the related research fields.

The purpose of our research is to gain a deeper understanding of the adsorption characteristics of CH_4_ in coal pores and to target the selection of relevant measures to enhance the CBM recovery ratio and to prevent coal mine gas disasters. Due to the pore size difference of micropores, it is of great importance to clarify the gas adsorption difference under different pore size conditions for the later adoption of corresponding measures to increase coalbed methane production. In our research, a simplified coal pore structure model with different pore sizes was constructed based on graphene sheets to study the effect of pore size on gas adsorption. The influence mechanism of pore size on gas adsorption characteristics was revealed by analyzing the isothermal adsorption curve, equivalent adsorption heat, and adsorption selectivity. Therefore, the results of this study have important theoretical and practical significance for exploring the occurrence state of coal pore gas at different scales and the mechanism of competitive adsorption difference to achieve efficient methane extraction.

## 2. Simulation Details

### 2.1. Slit Model Construction

Because coal is heterogeneous, the corresponding pore-structure model in the coal matrix has many morphological structures, including slits, cylinders, wedges, and bottlenecks, but generally, the proportion of slits is higher [31]. There are pore structures with different pore size distributions in coal. According to the IUPAC classification standard, the pore structure of coal can be divided into large pores (>50 nm), medium pores (2–50 nm), and micropores (<2 nm) [32]. The existing studies have confirmed that the gas adsorption and diffusion characteristics of pores can effectively reflect the interaction of the gas–solid interface at the micro level by using the graphene sheet–simplified coal pore-structure model. The slit model used in this study was based on the graphite structure (graphite.msi) provided by Materials Studio 8.0 (Accelrys, San Diego, CA, USA). Firstly, the symmetry was eliminated by making P1, and then a layer of graphite atoms was deleted and the cell structure was reconstructed. The cell was transformed into a cube structure of 90° × 90° × 90°. After obtaining the single-layer graphite structure, a supercell structure of 10 × 12 × 1 was constructed by the build-crystal–supercell functions, and the obtained graphene sheet size was 20.6 × 24.6 Å^2^. Since the coal pore wall has a certain thickness, in this study, the constructed supercell structure was operated along the z direction by the cleave surface, and one Å vacuum layer was added between each layer of graphene sheets by the constructing-layers operation to obtain a relatively stable pore wall structure. After obtaining the single-layer pore wall, the thickness of the vacuum layer between the graphene sheets was adjusted to 10 Å (1 nm), 20 Å (2 nm), and 40 Å (4 nm) through the build-layer function. The size of the model was 20.6 × 24.6 × 21.4/31.4/51.4 Å^3^, and the angle of the model was α = β = γ = 90°. The structural model is shown in Figure 1.

### 2.2. Simulation Details

In this study, the calculation and analysis of the molecular simulation were all realized using Materials Studio 8.0. The adsorption parameters of CH_4_ and CO_2_ in coal pores were acquired by the adsorption isotherm and fixed-pressure tasks in the sorption module. The temperature set by the simulation was 293.15 K, and the pressure range was 0–10 MPa. The pressure value selected in the simulation process was the fugacity value. In this study, Peng–Robinson realized the conversion between pressure and fugacity [33]. The relationship between pressure and the fugacity coefficient of the single-component gas and the binary-mixture gas is shown in Figure 2 and Figure 3. The force field in the adsorption process is the COMPASS force field, which is often used to simulate organic and inorganic molecules and can accurately predict the properties of molecules and polymers [34,35]. At the same time, many researchers have found that the COMPASS force field can be used to simulate the adsorption and diffusion properties of small gas molecules in macromolecules such as coal and shale.

The relevant parameters in the adsorption process were as follows: firstly, the coal pore structure and gas molecules were optimized by the geometry optimization task. Then, through the annealing simulation, ten cycles were performed with temperatures ranging from 300–600 K, which were calculated based on NVT. The dynamics tasks were conducted by NPT with the time of 1000 ps and with the time step of 1 fs, and when the energies were small enough, we could judge when the system had reached equilibrium. For methane and carbon dioxide adsorption in coal pores, the adsorption process was simulated by the GCMC method, which was realized mainly by the adsorption isotherm and fixed pressure tasks in the Sorption module. The number of adsorption equilibrium steps was 1 × 10^6^ steps, and the number of production steps was 2 × 10^6^ steps. The adsorption pressure was between 1–10 MPa, and the temperature was set to 293.15 K. The configuration calculation method was the metropolis method, the charge calculation method was forcefield assignment, the electrostatic calculation method was Ewald, and the accuracy was 1 × 10^−4^ kJ/mol. The van der Waals interaction calculation method was atom-based, and the truncation radius was set to 15.5 Å. Finally, the frames’ adsorbed CH_4_ and CO_2_ were simulated under NVT to obtain the gas competitive sorption and moving law in the coal pores.

### 2.3. Gas Adsorption Capacity and Adsorption Selectivity

The adsorption capacity in Materials Studio 8.0 is expressed as average molecules/cell, and the adsorption capacity is described generally as mmol/g in the experimental process. The unit conversion was performed with Equation (1) to make comparisons with the results in existing research:(1)$$uptake(\mathrm{mmol}/g)=\frac{Loading\;molecules}{M_{cell}(g/\mathrm{mol})}\times 1000$$

In the formula, *M_cell_* represents the adsorbent’s relative molecular mass in the unit cell.

When studying the competitive adsorption differences of multicomponent gases, it is necessary to introduce the selective adsorption coefficient to describe the separation characteristics of two different components during the competitive adsorption process. This is shown in Equation (2), which represents the mole ratio of the adsorbed phase and the bulk phase.
(2)$$S_{a/b}=\frac{xa/x_{b}}{ya/y_{b}}$$

In the formula, $x_{{}^{{}_{a}}}$($x_{b}$) represents the mole fraction of the adsorbed phase, $y_{{}^{{}_{a}}}(y_{b})$ represents the mole fraction of the bulk phase (a represents CO_2_ and b represents CH_4_). When $S_{a/b}$ > 1, it indicates that the competitive adsorption capacity of CO_2_ is more potent than that of CH_4_. The larger the $S_{a/b}$ is, the better the separation effect of the mixed gas. On the contrary, when $S_{a/b}$ < 1, CH_4_ has a more substantial competitive adsorption capacity than CO_2_. When $S_{a/b}$ = 1, the competitive adsorption capacities of the two gases are similar.

## 3. Results and Discussion

### 3.1. Adsorption Isotherms

The adsorption capacity variation under different molar ratios of CH_4_ and CO_2_ was obtained by Langmuir fitting to explore the character of gas competitive adsorption under different pore sizes. The selected molar ratios were 0.9:0.1, 0.7:0.3, 0.5:0.5, 0.3:0.7, and 0.1:0.9. At the same time, we also explored the adsorption capacity of CH_4_ and CO_2_ in the pores. The adsorption temperature was 293.15 K and the pressure range was 0–10 MPa. Figure 4 is the adsorption isotherm of the single-component gas in the pores, and Figure 5 shows the gas’s isothermal adsorption curve with different molar ratios. In the study of competitive adsorption of binary gases, due to the obvious competitive adsorption advantage of CO_2_, the adsorption amount of CH_4_ in binary mixtures is low, which is lower than 3 mmol/g. Therefore, the adsorption data of CH_4_ are not fitted in this study, and the curve only represents the changing trend of adsorption amount with pressure. The adsorption amount of CH_4_ is shown in Figure 6.

As shown in Figure 4, in the adsorption process of a single-component gas, the adsorption amount of CO_2_ under various conditions was significantly higher than that of CH_4_. In addition, the corresponding gas adsorption capacity was positively correlated with the pore size. Compared with the changing trend of the isothermal adsorption curve of CH_4_, the adsorption of CO_2_ reached saturation under low-pressure conditions. In Figure 5, except for CH_4_: CO_2_ = 0.9:0.1, the corresponding saturated CO_2_ adsorption capacities at the pore sizes of 1 nm, 2 nm, and 4 nm under other molar ratios were about 2.5 mmol/g, 7.5 mmol/g, and 20 mmol/g, respectively, showing that CO_2_ had apparent advantages in competitive adsorption, which was consistent with the existing research results [36,37]. The results of isothermal adsorption show that, with the increase in pore size, the competitive adsorption advantage of CO_2_ is gradually apparent. Only when the proportion of CO_2_ is deficient, such as 0.1, is the adsorption capacity of the two gases similar in the corresponding 1-nm and 2-nm pore sizes. Still, the adsorption capacity in the 4-nm pore is significantly different. It can be inferred that a higher CO_2_ concentration should be selected for competitive adsorption, and the competitive advantage of a low concentration of CO_2_ in ultramicropores is not obvious. Figure 6 shows the change in the CH_4_ adsorption amount in a binary gas under different molar ratios. Obviously, with the increase in the proportion of CO_2_ in the mixed gas, the CH_4_ adsorption amount shows a decreasing trend. When the mole ratio of CO_2_ was more than 0.5, the corresponding maximum adsorption capacity of CH_4_ was lower than 0.2 mmol/g. At the same time, when the mole ratio of CO_2_ was more than 0.7, the difference in CH_4_ adsorption amount under different apertures was not apparent. It can be inferred that 50–70% CO_2_ can be injected at a relatively low pressure to achieve effective displacement of methane in different temperatures to achieve the ideal CH_4_ displacement effect at a low cost.

### 3.2. Adsorption Selectivity

To determine the competitive adsorption capacity of CH_4_ and CO_2_ in coal pores, Figure 7 shows the relationship between the adsorption selectivity of the binary system and pressure at 298.15 K with different volume–mole fractions. In Figure 7, as the proportion of CO_2_ decreases, the corresponding S_CH__4/CO__2_ shows a decreasing trend. When the proportion of CO_2_ is higher than 0.5, the corresponding selective adsorption coefficient at the 2-nm pore size is larger, and the competitive adsorption advantage of CO_2_ at the 1-nm pore size is not apparent due to the small free-activity range. Due to the relatively sizeable mobile space of gas in the 2-nm coal pore, when the mole ratio of CO_2_ is less than 0.5, the competitive adsorption advantage of CO_2_ is more prominent. When the pore size is 4 nm, due to the extensive range of free space, the adsorption capacity of gas on both sides of the pore wall is enhanced, and finally, the competitive adsorption characteristics of gas in the pore are not obvious compared with the 2-nm pore. By comparing the competitive adsorption advantage of CO_2_ under different pore sizes and different molar ratios, it can be found that when the pore pressure is greater than 6 MPa, the competitive adsorption advantage of CO_2_ begins to decrease. At the same time, the competitive adsorption advantage decreases with the increase in CO_2_ concentration, which is also consistent with the isothermal adsorption law. The above research results are consistent with those of Hang et al.; that is, the higher CO_2_ concentration in the mixed gas corresponds to a smaller adsorption selectivity coefficient [38]. Through comprehensive analysis of the change law of the selective adsorption coefficient, it can be found that when the proportion of CO_2_ is less than 0.5, the highest selective adsorption coefficient appears at 2 MPa. These results show that relatively low pressure and CO_2_ concentration are more conducive to CH_4_ displacement.

### 3.3. Isosteric Heat of Adsorption

The variation law of the isothermal heat of adsorption of single-component and binary-mixture gases in pores is shown in Figure 8. The isothermal heat of adsorption is an important parameter for describing the interaction between gas molecules and adsorbents. In addition, it is also an important parameter to characterize the energy variation law of the adsorption system [39,40]. The calculation method of isothermal adsorption heat is shown in Equation (3):(3)$$(\ln P{)}_{N}=\frac{Q_{st}}{R}\times \frac{1}{T}+C$$

In the formula, R is the ideal gas constant, m^3^·Pa·mol^−1^·K^−1^; *P* is the pressure, Pa; N is the quantity of adsorbate (gas loading); T is the temperature, K; $Q_{st}$ is the isosteric heat of adsorption (kJ/mol); and C is the constant [41].

As shown in Figure 8, the change trends in the isosteric heat of adsorption in single- and binary-mixture components are the same. At the same time, according to the research results of Xiang et al., when the equivalent adsorption heat is less than 42 kJ/mol (10.03 kcal/mol), the adsorption types are all physical adsorption [42]. The isosteric heat of adsorption of CH_4_ in this study was about 4–7 kcal/mol, which is typical physical adsorption. In comparison, the isosteric heat of adsorption of CO_2_ was greater than 10 kcal/mol, which is chemical adsorption. This research result is consistent with Gao et al. [43]. Under different pore sizes, there is no noticeable difference in the isosteric heat of adsorption under different pressure conditions. When the pore size was 2 nm, the isosteric heat of adsorption of CH_4_ at the molar ratio of 0.1 under 2 MPa was lower than that of pure methane under this condition, indicating that relatively low pressure and low CO_2_ concentration are beneficial to the desorption of CH_4_, which is consistent with the results of 3.1 and 3.2. Similarly, when the pore size was 4 nm, the isosteric heat of adsorption was lower than that of pure methane only when CH_4_:CO_2_ = 0.9:0.1. With the increase in pore size, the isosteric heat of adsorption of the corresponding gas gradually decreased, indicating that with the increase in pore size, the adsorption capacity of pore wall to gas gradually decreases; that is, the amount of CH_4_ in the corresponding free space gradually increases, which is consistent with the increasing trend of adsorption capacity.

## 4. Conclusions

(1) In the adsorption process of a single-component gas, the adsorption amount of CO_2_ under various conditions was significantly higher than that of CH_4_, and the adsorption amount was positively correlated with the pore size. With the increase in pore size, the competitive adsorption advantage of CO_2_ is gradually apparent. Among them, a higher CO_2_ concentration should be selected in the competitive adsorption of CH_4_ in ultramicropores (1 nm). In micropores, 50–70% CO_2_ injection under relatively low pressure can achieve an ideal yield-increase effect.

(2) By comparing the variation of the selective adsorption coefficient under different molar ratios, it can be seen that, when the pore pressure is greater than 6 MPa, the competitive adsorption advantage of CO_2_ begins to decrease. At the same time, the competitive adsorption advantage decreases with the increase in CO_2_ concentration. When the molar ratio of CO_2_ in the binary mixture is more than 0.5, the corresponding selective adsorption coefficient is the highest at 2 MPa. Lower pressure and lower CO_2_ concentration are favorable for CH_4_ desorption.

(3) With increased pore size, the isosteric heat of adsorption of the corresponding gas gradually decreases. The adsorption capacity of the pore wall to gas gradually weakens, the amount of CH_4_ in the free space gradually increases, and the desorption effect of CH_4_ gradually decreases significantly, which is more conducive to the production increase in CH_4_.

(4) Due to the study of competitive adsorption differences in different pore sizes in this paper, to simplify the calculation of the coal pore model, only graphene was used to simplify the pore model and the pore size range of the study was micropores. In future research, the pore structure of the real coal body will be further optimized, and the research scope of pore size will be further expanded.

## Figures and Tables

**Figure 1 molecules-27-01594-f001:**
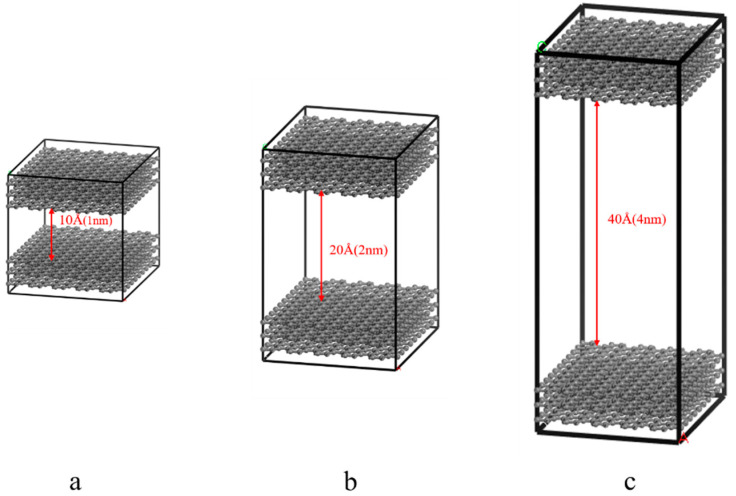
Pore structure model of coal ((**a**): 1 nm; (**b**): 2 nm; (**c**): 4 nm).

**Figure 2 molecules-27-01594-f002:**
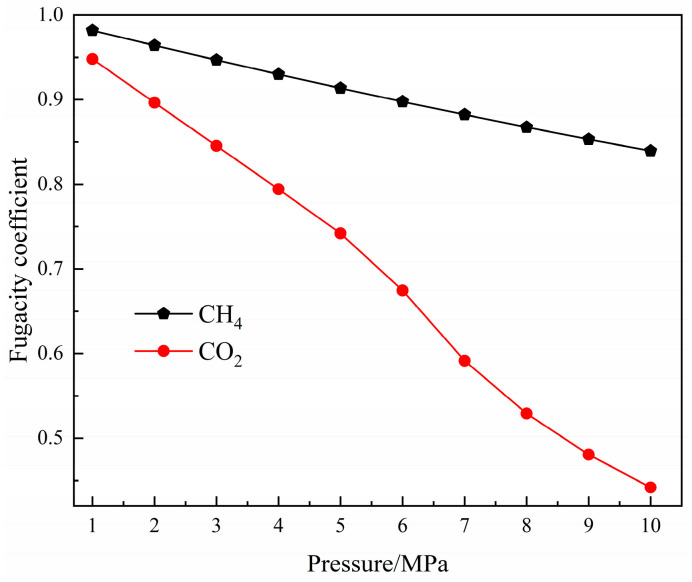
The relationship between the fugacity coefficient of gas and its pressure.

**Figure 3 molecules-27-01594-f003:**
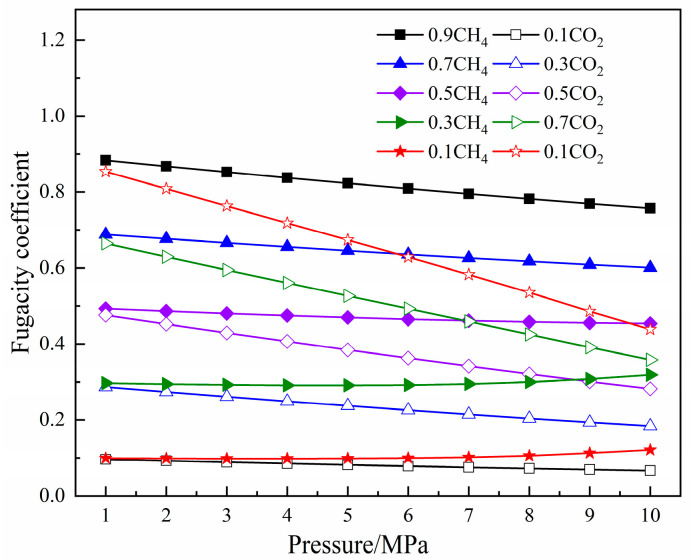
The relationship between the fugacity coefficient of binary components and pressure.

**Figure 4 molecules-27-01594-f004:**
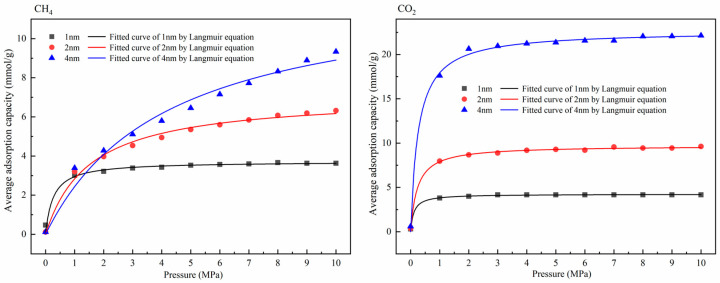
Adsorption isotherms of CH_4_ and CO_2_ at 293.15 K.

**Figure 5 molecules-27-01594-f005:**
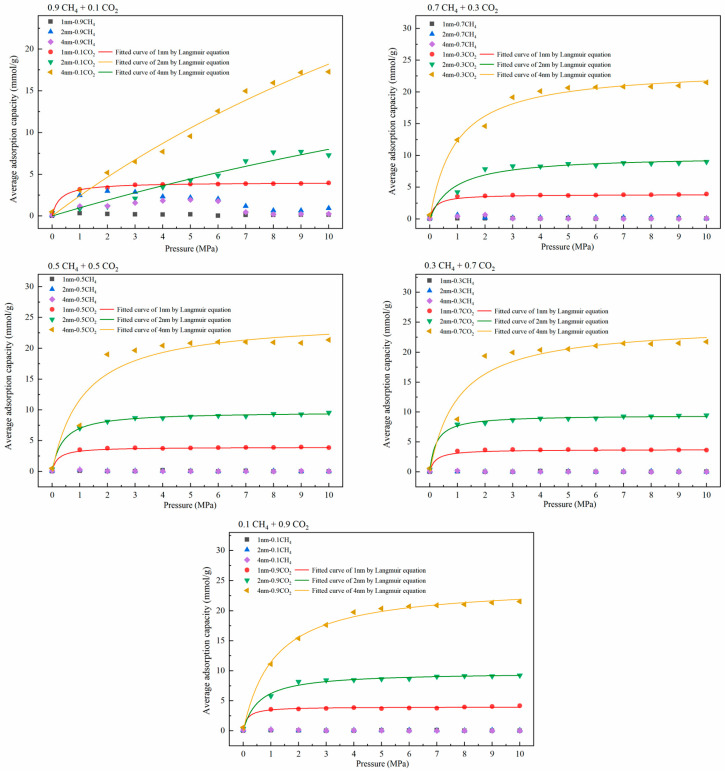
Adsorption isotherms of different ratios of CH_4_/CO_2_ at 293.15 K.

**Figure 6 molecules-27-01594-f006:**
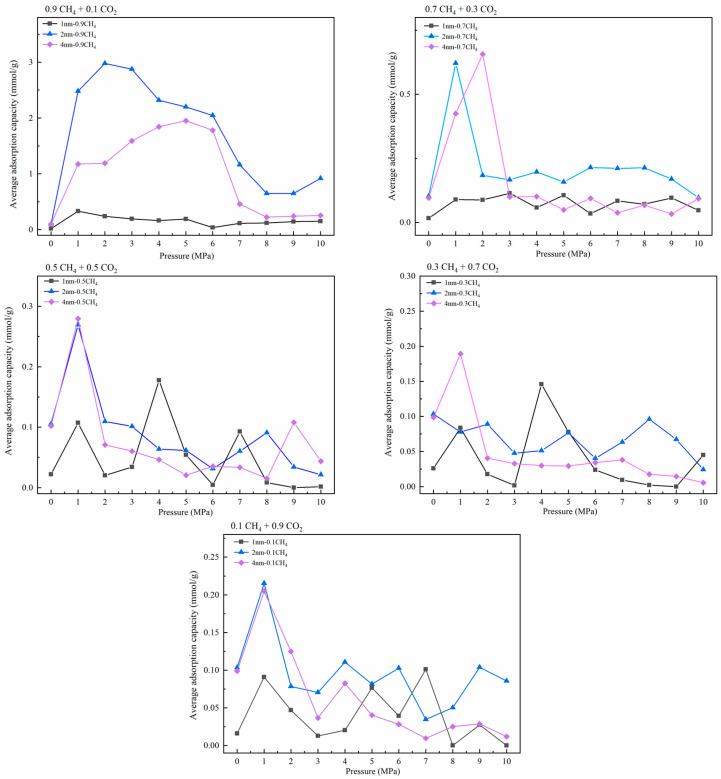
Adsorption isotherms of different ratios of CH_4_ at 293.15 K.

**Figure 7 molecules-27-01594-f007:**
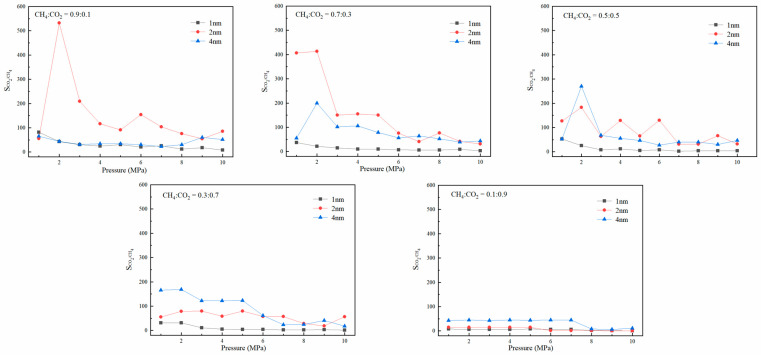
Adsorption selectivity of different ratios of CH_4_/CO_2_ at 293.15 K.

**Figure 8 molecules-27-01594-f008:**
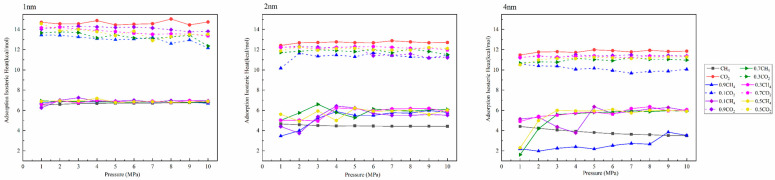
Isosteric heat of adsorption of CH_4_, CO_2_, and different ratios of CH_4_/CO_2_ at 293.15 K.

## Data Availability

Not applicable.

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
