# Peer review of "Molecular Simulation on Competitive Adsorption Differences of Gas with Different Pore Sizes in Coal"

_molecules, 2022, doi:10.3390/molecules27051594_

Round 1

Reviewer 1 Report

The manuscript titled, "Molecular simulation on competitive adsorption differences of  gas with different pore sizes in coal" reported by Han et al., has accumulated and presented the scientific problem meticulously. The manuscript can be accepted in its current form.

Author Response

Thank you for your professional review, we will continue to strive to publish more excellent articles.

Reviewer 2 Report

REVIEW

Remarks & Suggestions to Authors

In this study, the Authors present a “Grand Canonical Monte Carlo” (GCMC) simulation treatment aiming to obtain detailed information with regard to the gas adsorption in coal-pores with different pore sizes. To realize their target the Authors employed a simplified coal pore structure model with different pore sizes based actually on graphene sheets. More specifically, the main goal of this research mentioned by the Authors is,to gain a deeper understanding of the adsorption characteristics of CH4 in coal pores and …… to enhance the CBM (coalbed methane) recovery radio  and prevent coal mine gas disastersFrom this point of view and according to the relevant literature on the field, the aforementioned topic is undoubtedly interesting and the simulation technique used appropriate for such kind of investigations in general. On the other hand, evaluating the scientific merit of the present study based on my detailed examination the Sections one by one of the MS, I have to mention that prior any final decision the writing of the MS can be more improved conceptually since a number of issues and considerations should be clearly justified (see below).

A. The Authors are strongly requested to check English systematically (-by a native speaker or by the Language service of the Journal) in the revised MS before re-submission.

B. Please rewrite the abstract Section and clarify the core research findings.

C. Improve Sec 1. Introduction & Update the references accordingly.

C1 on p. 2, lin. 46 “Many researchers have … on experimental measurement and.. methods”.[xx]. Please provide restric. information regarding the exper. measurement & other methods as well insert relevant ref(s).

C2 on p. 2, lin. 50 “At the same time, the methane … is difficult … extracted”. A few words for the main reason(s) behind this problem are needed here.

C3 on p. 2, lin. 52-54 “Song et al. taking… object, studied the …adsorption… of CH4 and CO2 …experiments and sim. methods.[xx]. Please insert the relevant ref(s) to the above study, since ref. [12] in ref. listing corresponds to other Authors than Song et al. .

D. In the last parag. of Introduction on p.2 lin. 83-91, “Due to the …. on graphene sheets to study the effect of pore size on gas adsorption. The influence mechanism of pore size on gas adsorption characteristics was revealed by analyzing the isothermal adsorption curve, equivalent adsorption …. and prevent coal mine gas disasters”, the Authors provide information regarding the main purpose of their simulation study. It’s however not clear from the corresponding prepared text the substantial differential of the above mentioned target and methodology used from those reported in the previous similar studies [see cited refs 16 & 17]. To this point, I have to note that, the Authors are strongly advised to provide useful information about this consideration and to critically discuss (in
the revised MS, Sec. Results and Discussion) the main conclusions reported in those previous simulation studies with their predictions. I feel that such a comparison
might be of particular interest for the anonymous reader in general.

F. About Sec. 2. Simulation details

    2.1. Slit model construction

Concerning the Slit model construction on p.3, lin. 106-108, “The thickness of the vacuum layer between graphene is adjusted to 1 nm, 2 nm, and 4 nm through the build layer. The size of the model is 20.6*24.6*21.4/31.4/51.4 Å, and the angle of the model is α = β = γ = 90. The structural model is shown in Figure 1.”, the Authors provide information about the geometrical characteristics of the model used. However I feel that such a description as the above one seems to be incomplete and the Authors are strongly requested to explain the constructed model elaborately. Concretely speaking, the calculation used to estimate the size of the model must be further clarified. Note in addition that the caption in Fig .1 should be improved.

2.2. Simulation details                                                                                                                                                                                                                       

By inspecting sub-Sect. 2.2 Simulation details,

[on p. 3 & lin. 111-130,“ In this study, the calculation and analysis of molecular simulation are all realized by 112 Material Studio 8.0. The adsorption parameters of CH4 and CO2 in coal pores ……which is often used to simulate organic and inorganic molecules and can accurately predict the properties of molecules and polymers [25, 26]. At the same time, many researchers   …..   interaction is Ewald. The sampling interval   …… is 50 steps. The van der Waals … is Atom based, and the cut-off radius is  15.5 Å. “],

it’s seen that the above mentioned sub-Section provides information about the simulation technique used (Grand Canonical Monte Carlo, GCMC) to accomplish the target of this treatment. From this point of view and according to the relevant literature, I have to state that I agree with the above, that is, that the usage of the GCMC technique upon this particular problem here is indispensable. Nevertheless, I found the simulation details provided via the corresponding text                   to be unjustifiably quite limited and in addition not professionally prepared. It’s obvious therefore that to make the reviewing process easier and promptly, the Authors are requested to improve sub-Sect. 2.2 Simulation details substantially. To make this possible, the Authors are strongly advised to do the following. Concretely, they have to provide in the revised text detailed information about all the necessary and basic stages – one by one- of the GCMC simulation-execution of each molecular system studied, starting as usually from the preparation of the initial configuration of any system into the slit model and afterwards by describing the next stages of the process until the estimation of the properties of interest. To this point, the Authors are suggested to look back in the literature, namely, in specific reviews and mainly in GCMC simulation papers with great citation index, to obtain information how to manage the above request successfully. For instance, I could mention the following indicative publications:

a) Review of Molecular Simulation Method for Gas Adsorption/desorption and Diffusion in Shale Matrixwritten by, H. Wang et al, Journal of Thermal Science (Springer) vol. 28, p. 1–16 (2019);

b) “Molecular simulation of gas adsorption in shale nanopores: A critical review Wang et al. Renewable and Sustainable Energy Reviews vol. 149, (2021), 111391;https://doi.org/10.1016/j.rser.2021.111391

c)Adsorption of N2, CH4, CO and CO2 gases in single walled carbon nanotubes: A combined experimental and Monte Carlo molecular simulation study”, written by GP Lithoxoos, et al., The Journal of Supercritical Fluids 55 (2), 510-523, (2010)

As seen specifically from the above ref.(c), this previous experimental & GCMC simulation study provides adsorption results for CH4 and CO2 gases in SWNTs at about similar thermodynamic conditions to the present simulation. Therefore, the Authors are strongly advised to cite all the aforementioned GCMC sim. works in Introduction of the revised MS appropriately (by introducing a short text prepared properly), and certainly to provide restricted discussion based upon the comparison of the results obtained from both simulations (the present one and the previous ref. c ).

To conclude, this paper is probably publishable, but should be reviewed again (major revision)   in a re-submitted revised form before it is accepted for publication.

Reviewer 3 Report

Paper entitled “Molecular simulation on competitive adsorption differences of 2 gas with different pore sizes in coal” meets the necessary standards for publication in this journal.

Please check the entire manuscript carefully for eventual typographical errors.

Consider necesar sa se explice abrevierea GCMC din abstract.

Attention when writing references. They are not unitary.

Final Conclusion: The paper meets the necessary standards for publication.

Author Response

Point1: Please check the entire manuscript carefully for eventual typographical errors.

Response: Thank you for your comments. We have checked the entire manuscript carefully for eventual typographical errors.

Ponit2:Consider necesar sa se explice abrevierea GCMC din abstract.

Response: Thank you for your comments. We have add the shorthand of GCMC (Grand Canonical Monte Carlo) in abstract.

Ponit3:Attention when writing references. They are not unitary.

Response: Thank you for your comments. We have checked all references in our  manuscript carefully and revised the wrong  form references.

Round 2

Reviewer 2 Report

Final Review

After re-evalation of the revised MS, I found the new MS acceptable for publication